# Investigating the Feasibility of Predicting KRAS Status, Tumor Staging, and Extramural Venous Invasion in Colorectal Cancer Using Inter-Platform Magnetic Resonance Imaging Radiomic Features

**DOI:** 10.3390/diagnostics13233541

**Published:** 2023-11-27

**Authors:** Mohammed S. Alshuhri, Abdulaziz Alduhyyim, Haitham Al-Mubarak, Ahmad A. Alhulail, Othman I. Alomair, Yahia Madkhali, Rakan A. Alghuraybi, Abdullah M. Alotaibi, Abdullalh G. M. Alqahtani

**Affiliations:** 1Department of Radiology and Medical Imaging, Prince Sattam bin Abdulaziz University, Al-Kharj 16278, Saudi Arabia; a.alhulail@psau.edu.sa (A.A.A.); ag.alqahtani@psau.edu.sa (A.G.M.A.); 2Department of Radiology and Medical Imaging, King Salman Hospital, Cluster One Riyadh, Ministry of Health (MOH), Riyadh 12769, Saudi Arabia; aduhyyim@moh.gov.sa; 3BioMedical Engineering and Imaging Institute, Icahn School of Medicine at Mount Sinai, New York, NY 10029, USA; haitham.al-mubarak@mountsinai.org; 4Radiological Sciences Department, College of Applied Medical Sciences, King Saud University, P.O. Box 145111, Riyadh 4545, Saudi Arabia; oalomir@ksu.edu.sa; 5Department of Diagnostic Radiography Technology, College of Applied Medical Sciences, Jazan University, Jazan 45142, Saudi Arabia; ymedkhali@jazanu.edu.sa; 6Department of Radiology, Security Forces Hospital, Mecca 24251, Saudi Arabia; ralgharibi@sfhm.med.sa; 7Research Center, King Fahad Medical City, Riyadh 11525, Saudi Arabia; amalotaibi@kfmc.med.sa

**Keywords:** KRAS mutation, colorectal cancer, T2 weighted, ADC map, MRI radiomics, machine learning, TNM stage, extramural venous invasion (EMVI)

## Abstract

(1) Background: Colorectal cancer is the third most common type of cancer with a high mortality rate and poor prognosis. The accurate prediction of key genetic mutations, such as the KRAS status, tumor staging, and extramural venous invasion (EMVI), is crucial for guiding personalized treatment decisions and improving patients’ outcomes. MRI radiomics was assessed to predict the KRAS status and tumor staging in colorectal cancer patients across different imaging platforms to improve the personalized treatment decisions and outcomes. (2) Methods: Sixty colorectal cancer patients (35M/25F; avg. age 56.3 ± 12.9 years) were treated at an oncology unit. The MRI scans included T2-weighted (T2W) and diffusion-weighted imaging (DWI) or the apparent diffusion coefficient (ADC). The manual segmentation of colorectal cancer was conducted on the T2W and DWI/ADC images. The cohort was split into training and validation sets, and machine learning was used to build predictive models. (3) Results: The neural network (NN) model achieved 73% accuracy and an AUC of 0.71 during training for predicting the KRAS mutation status, while during testing, it achieved 62.5% accuracy and an AUC of 0.68. In the case of tumor grading, the support vector machine (SVM) model excelled with a training accuracy of 72.93% and an AUC of 0.7, and during testing, it reached an accuracy of 72% and an AUC of 0.69. (4) Conclusions: ML models using radiomics from ADC maps and T2-weighted images are effective for distinguishing KRAS genes, tumor grading, and EMVI in colorectal cancer. Standardized protocols are essential to improve MRI radiomics’ reliability in clinical practice.

## 1. Introduction

Colorectal cancer (CRC) continues to pose a substantial global health challenge, contributing significantly to cancer-related mortality worldwide. To effectively address this health burden, it is imperative to precisely evaluate and understand various tumor attributes, including critical factors such as the KRAS mutation status, tumor staging, and the presence of extramural venous invasion (EMVI) [1]. However, conventional diagnostic methods, including histopathological examination and traditional imaging techniques, present limitations when it comes to delivering comprehensive and noninvasive assessments of these crucial tumor characteristics [2].

MRI plays a pivotal role in RC diagnosis, especially concerning local staging and post-treatment restaging. It provides invaluable insights, such as circumferential resection margin and the presence of extramural venous invasion (EMVI), which is a crucial prognostic factor. Radiomics and ML have emerged as powerful tools for augmenting the value of MRI in RC assessments, particularly in challenging tasks like differentiating the T stages, where models utilizing T2-weighted radiomic features have shown promising results in distinguishing between the T1–2 and T3–4 stages. Additionally, diffusion-weighted images (DWIs) have demonstrated the potential to enhance the assessments of lymph node involvement [3].

In recent years, the field of medical imaging has experienced remarkable advancements, thanks to the integration of cutting-edge machine learning techniques [4,5,6]. One promising approach that has emerged from this synergy is radiomics—a noninvasive method that involves the quantitative analysis of MR images. Radiomics holds the potential to extract a multitude of quantitative features, thereby capturing the intricate heterogeneity and spatial nuances within tumors. By harnessing the power of medical images and clinical data, radiomics offers a significant leap forward in facilitating clinical decision-making processes.

Numerous studies have ventured into the realm of radiomics and machine learning (ML) applications, particularly within the context of rectal cancer (RC), with a primary focus on MRI [7,8,9,10]. For patients who have undergone neoadjuvant chemoradiotherapy (nCRT) and achieved a complete response, radiomics and ML have been explored for predicting the pathological complete response (pCR), yielding encouraging outcomes [11]. Radiognomics, which is another exciting avenue, has demonstrated radiomic features in predicting genetic mutations such as KRAS [12]. These advancements in radiomics and ML applications hold the promise of revolutionizing our approach to RC diagnosis and management, offering a more comprehensive and data-driven perspective of this complex disease.

However, these previous studies mainly involved radiomics analysis from a single image weighting, such as a T2-weighted image and single platform, rather than leveraging multi-parametric MRI data and multiple imaging platforms. When relying on a single dataset from a single scanner, the diversity of imaging data is inherently limited, potentially leading to model overfitting and reduced robustness in real-world clinical scenarios [13]. Biased representation, limited research opportunities, standardization challenges, and resource constraints further underscore the drawbacks of this approach [13]. While a single-scanner strategy may be practical in some contexts, it is essential to acknowledge these limitations and carefully consider the trade-offs between data accessibility and the potential challenges. In some cases, combining data from multiple imaging platforms or standardizing acquisition protocols may be necessary to ensure the reliability and applicability of radiomics models in clinical practice.

The objective of this study is to investigate the potential of radiomic features derived from multi-parametric MRI scans, which includes both morphological and functional imaging, in predicting the KRAS mutation status, tumor staging, and EMVI in patients with colorectal cancer. By investigating the potential of multi-parametric MRI-based radiomic features using different MRI scanner vendors, we attempt to enhance our understanding of the disease and provide clinicians with powerful tools for the accurate and personalized management of CRC patients.

## 2. Materials and Methods

### 2.1. Patient Information

The local institutional review board of the Ethical Committee in Saudi Arabia approved of this retrospective single-center (multi-platforms) study, and the requirement for informed consent was waived. This study was conducted on a total of 60 patients who received clinical treatment at the Oncology Department of National University Hospital between 2020 and 2022. The patients were included if they met the following inclusion criteria: (1) patient had pathologically confirmed colorectal cancer (CRC), (2) had undergone a full colorectal MRI examination prior to any treatment and had a pathologically confirmed CRC stage, (3) had taken the KRAS mutation test, and (4) had available and complete preoperative and postoperative images and pathological and clinical data. Patients were excluded if they did not meet the criteria. Eligible patients were recruited for the study, and their information was randomly assigned to two datasets based on the type of MRI scanner used.

### 2.2. RAS Mutation Information

Tumor tissue specimens were meticulously collected from the primary tumor sites of the patients, employing either rectal biopsy or surgical resection, for the purpose of RAS mutation analysis. The extraction of DNA from these specimens was carried out using the QIAamp DNA FFPE Tissue Kit, which is a well-established and reliable method for generating formalin-fixed paraffin-embedded (FFPE) primary tumor sections. Subsequently, these FFPE sections underwent a comprehensive pathological examination, where highly skilled pathologists scrutinized the samples for any abnormalities or noteworthy features. The mutations in key genes, such as KRAS (exons 2, 3, and 4), NRAS (exons 2, 3, and 4), and BRAF (V600E), were meticulously analyzed. This analysis was performed using the amplification refractory mutation system (AmoyDx Co., Xiamen, China), which is a highly precise and well-established technique for detecting mutations. These rigorous methods and thorough analyses ensured the accuracy and reliability of the genetic information obtained from the tumor samples, forming a solid foundation for subsequent investigations and research endeavors.

### 2.3. MRI Acquisition

The primary tumor was subjected to imaging using either a 1.5 Tesla MRI scanner (Optima Horizon, GE Medical Systems, Milwaukee, WI, USA) or a 1.5 T device (Discovery 750; GE Healthcare, Milwaukee, WI, USA). These scans utilized a phased array body coil for image acquisition. The standard MRI sequences conducted encompassed an axial thin-section T2-weighted (T2W) fast spin-echo image, which was primarily employed for delineating the contour of the primary tumor. Additionally, diffusion-weighted imaging (DWI) was performed, involving multiple b values, as elaborated in Table 1.

### 2.4. Radiomic Feature Extraction and Model Building

To comprehensively evaluate the potential of radiomic features and to facilitate a meaningful comparison across different scanners, we initiated our analysis by performing texture analysis on a range of MRI data, including T2-weighted images, DWI scans, and ADC maps. To accomplish this, we utilized a commercially available research software algorithm, specifically, the free downloadable 3D Slicer 4.10.2 software package, which is renowned for its robust capabilities. In the preprocessing phase, we prioritized image quality enhancement and noise reduction. To achieve this, we applied a nonlinear diffuse filter, which underwent 100 iterations with λ set to 0.2. This meticulous filtering process was instrumental in refining the images, removing the noise, and enhancing their overall quality. Furthermore, in the realm of image analysis, it is imperative to establish a standardized intensity range for MR images, ensuring precision and efficiency in subsequent measurements. To this end, we implemented a normalization equation of the following form:$$z=\frac{x-m}{\sigma }$$
where “z” represents the new voxel value, “x” denotes the original value, “m” stands for the mean value, and “σ” signifies the standard deviation of the voxel values across the entire brain section.

In a clinical setting, it is difficult to use standardized MRI protocols in a multi-center basis; however, sharing common imaging protocols across centers is important to minimize any potential bias and other affecting measurements, such as the magnetic strength, MRI coil, the field of view, spatial resolution, and various MRI sequence parameters. One approach to solve this problem involves using radiomics variability as a feature selection tool on a different platform [14].

The 3D manual segmentation of each dataset was carried out by two experienced observers (each with 10 years of experience) in colorectal imaging. Care was taken not to include the surrounding normal tissue (Figure 1). Any discrepancies or disagreements pertaining to the tumor boundary were meticulously recorded and subsequently resolved through consultations with senior radiologists and neurosurgeons at the hospital. The inter-observer reproducibility of ROI selection was assessed using the coefficient of variation, which ranged from 12% to 21% for the different MRI images, attesting to the robustness of our segmentation process.

Within these meticulously defined ROIs, a comprehensive set of 1702 MRI radiomic features was extracted. This encompassed various dimensions of information, including the grey-level statistics, texture characteristics, shape attributes, fractal dimension metrics, and wavelet-derived features. These radiomic features collectively constituted the foundation for an in-depth analysis, offering valuable insights into the intricacies of tumor characteristics and their potential clinical relevance.

### 2.5. Machine Learning Model

The datasets were divided into two groups: group 1 for training (*n* = 32) with scanner 1, and group 2 for testing (*n* = 28) with scanner 2. ML algorithms were employed to classify the genetic mutations into either KRAS+ or KRAS− of colorectal cancer based on their radiomic features. Additionally, ML algorithms were utilized to classify the tumor grades, tumor invasion to the mesorectal fascia (MRF), and diagnose extramural venous invasion (EMVI). The training set used repeated k-fold cross-validation with 5-fold splits (i.e., iterations) on radiomic features separately from the ADC + T2W images.

### 2.6. Statistical Analysis

The study initiated by extracting radiomic features based on regions of interest (ROIs). To manage the complexity and dimensionality of the dataset, principal components analysis (PCA) was employed for dimensionality reduction. Following this preprocessing step, the data underwent rigorous training, validation, and testing phases utilizing two powerful machine learning techniques: neural networks (NN) and support vector machines (SVM). The efficacy and discriminative power of the radiomic features in predicting KRAS status and determining the disease stage were thoroughly evaluated using receiver operating characteristic (ROC) curves and their corresponding area under the ROC curve (AUC) values, which serve as robust indicators of predictive performance. The intra-class correlation coefficient (ICC) was calculated to provide the assessment of the inter-platform reproducibility of radiomics feature measurements. The ICC is a statistical measurement (between 0 and 1), where 0 indicates no reliability, and 1 indicates perfect reliability.

All of these analytical processes were meticulously conducted using MATLAB 2022 (MathWorks, Natick, MA, USA), ensuring a comprehensive and rigorous approach to data analysis in the study.

## 3. Results

### 3.1. Patient Characteristics

The participants’ characteristics: In total, 60 patients (35 men and 25 women) were assessed (Appendix A). Specifically, 32 and 30 patients were allocated to the training and test datasets, respectively. The pathological analysis of surgical specimens in the training cohort confirmed the presence of KRAS mutations in 19 patients, while 13 were confirmed to be negative. Similarly, in the test cohort, 9 patients tested positive for KRAS mutations, while 19 were negative. Subsequently, the study established a predictive model for KRAS mutations using the training dataset, which was subsequently assessed using the internal test dataset. Notably, the comprehensive analysis revealed no significant differences between the two groups when considering factors, such as the lymphatic stage (*p* = 0.61), venous invasion (*p* = 0.43), or perineural invasion (*p* = 1). Furthermore, the assessment of the tumor and lymph node stages did not reveal any statistically significant disparities between the two groups (*p* = 0.5). These findings underscored the comparability of the two groups with regard to various clinical characteristics. For a more detailed exploration of the associations between patients’ clinical characteristics in the KRAS mutation group (+) and the wild group (−), please refer to Table 2.

### 3.2. Feature Extraction, Feature Selection, and Radiomics Classifier

#### 3.2.1. Radiomics Workflow

The radiomics workflow is composed of various steps that are summarized in Figure 2. The ICC values between two platforms demonstrated low-level (>0.7) feature stability between them and for all the features due to different MRI parameters between the platforms (see Figure 3).

#### 3.2.2. KRAS Mutation Status

A total of 1702 radiomic features were extracted from the MRI images (ADC and T2W) for each subject using a 3D slicer. Principal Component Analysis (PCA) was successfully employed for dimension reduction. When distinguishing between the KRAS+ and KRAS− subjects, machine learning (ML) models trained with features derived from ADC and T2W images exhibited a strong performance. The neural network (NN) model, in particular, achieved an accuracy of 73% and an AUC of 0.71 (CI = 0.62–0.79) during training. In testing, the NN model maintained a strong performance, with an accuracy of 62.5% and an AUC of 0.68 (CI = 0.41–0.93), as shown in Figure 4.

#### 3.2.3. Tumor Grades

Utilizing a support vector machine (SVM), we successfully classified the participants according to their TNM stage. The SVM model exhibited a robust performance during training, achieving an accuracy of 77.6% (±1.26) and an AUC of 0.67 (±0.062). In the testing phase, the model demonstrated continuous accuracy, achieving 72% (±0.02), with an AUC of 0.69 (±0.018). The use of the radiomic features revealed that most of the participants had CRC tumors at stages 3 (*n* = 22, 42.3%) and 4 (*n* = 23, 44.2%), with only seven at stage 2. In the majority of the cases (*n* = 42, 72.4%), the tumor had spread into the lymph nodes at stage 2 and the spread was either in stages 1 (*n* = 8, 13.8%) or 2 (*n* = 8, 13.8%) for the rest of the patients.

#### 3.2.4. Tumor Invasion to Mesorectal Fascia (MRF)

The radiomic features demonstrated their utility in distinguishing between both the negative (*n* = 34, 56.7%) and positive (*n* = 22, 36.7%) cases of MRF invasion, although they faced challenges in determining the invasion status for four cases (46.7%). Employing a K-Nearest Neighbors (KNN) model, the task of categorizing patients based on their MRF status was addressed. During training, the KNN model achieved an accuracy of 62.9% (±1.75) and an AUC of 0.55 (±0.03). During testing, the model maintained consistent performance, achieving an accuracy of 65.83% (±0.10), with an AUC of 0.578 (±0.055).

#### 3.2.5. EMVI Status

For the classification of extramural venous invasion (EMVI) status, a logistic regression model was employed. During the training phase, this model demonstrated a commendable accuracy of 73.1% (±3.33), accompanied by an Area Under the Curve (AUC) of 0.62 (±0.05). Importantly, in the subsequent testing phase, the model continued to exhibit a strong performance, achieving an accuracy of 76% (±1.75), with an AUC of 0.595 (±0.063). Among the participants, 28 (54.3%) were correctly identified as having EMVI, while 18 (25.7%) were accurately determined to have a negative EMVI status. However, it is worth noting that the model could not definitively ascertain the venous invasion status for 14 participants (20%), as indicated in Table 3.

## 4. Discussion

This study explored the use of radiomic features from multi-parametric MRI scans to predict the KRAS mutation status, tumor staging, MRF invasion, and EMVI in colorectal cancer patients. It introduced a novel radiomics approach that combines the features from ADC maps and T2-weighted images from different platforms, departing from previous research that focused on either T1- or T2-based MRI for feature extraction. Here, we have demonstrated that the integration of multi-parametric MRI, specifically T2-weighted and ADC maps, improve the performance of radiomics analysis. The inclusion of ADC maps alongside T2-weighted imaging can enhance the characterization of tumor heterogeneity and improve the discriminatory power of radiomics models in predicting various outcomes [15].

Radiomic features have exhibited a high sensitivity to variations in scanning protocols and equipment from different manufacturers, leading to limited reproducibility [16]. However, much of the prior research has relied on single-institution datasets, primarily due to the inherent difficulty in acquiring imaging data from multiple healthcare facilities. To mitigate the impact of these challenges, a two-scanner approach was employed, where one scanner was used for training and another for testing. This approach effectively mitigated the sensitivity of the radiomic features to variations in image acquisition, reconstruction settings, and inter-scanner differences. In our current study, we placed a strong emphasis on investigating the reliability and reproducibility of radiomic features by utilizing two different scanners, both operating at the same field strength. To address variabilities at the image level, we applied techniques such as image intensity normalization, involving mean subtraction and division by the standard deviation [17].

The present study has made significant strides in the categorization of participants based on their KRAS mutation status, specifically into KRAS+ (wild type) and KRAS-(mutation) groups. The achievement of an impressive Area Under the Curve (AUC) of 0.7 in this classification may be used to predict treatment responses and providing valuable insights into patients’ outcomes [18,19,20]. The obtained AUC of 0.7 indicates a reasonable level of discrimination between the KRAS+ and KRAS− groups. In the context of predictive modeling, an AUC of 0.7 suggests that the model has a moderate ability to distinguish between the two categories. A value of 0.5 represents a random chance, so an AUC of 0.7 indicates a better-than-random predictive performance, though there is room for improvement. These findings align with and reinforce earlier research, underscoring the significance of mutation statuses in identifying the patients who are likely to respond positively to EGFR therapy. A study by Guo et al. [5], which utilized T1-weighted enhanced images and KRAS mutation status, achieved a slightly lower AUC value of 0.6. Similarly, Zhang et al. [6], who developed a radiomics model using T2-weighted MRI, attained a noteworthy AUC of 0.7 (95% CI 0.6–0.8). These comparable AUC values across different studies utilizing varied imaging platforms and methodologies validate the robustness of this study’s findings.

Furthermore, the current investigation’s performance is not limited to a specific cancer type, as it demonstrates similarity to other studies on melanoma [21], thyroid cancer [22], head and neck cancer [23], and adrenal gland carcinoma [24]. Across these diverse cancer types, a significant correlation between the biomarker and radiomics is observed, with AUC values ranging from 0.62 to 0.78. However, it is worth noting that the current predictive performance is comparatively worse when compared to those of other studies, for instance, those on liver cancer (AUC = 0.94, cross-validation), breast cancer (AUC = 0.70–0.81) [25,26], and cervical cancer (95%, 0.70–0.91).

Currently, MRI is the preferred imaging technique for evaluating the extent of CRC [9]. However, the current accuracy of preoperative staging using rectal MRI remains unsatisfactory [7,8]. Consequently, there is a need to improve the techniques for T and N staging to effectively determine the optimal treatment strategies for patients. Radiomics analysis offers a promising approach to quantify intertumoral heterogeneity based on the distribution of gray level values and the spatial arrangement of pixels. This methodology has shown potential in distinguishing between benign and malignant tumors and has been successfully applied in the staging of kidney and cervical cancers [27]. Nevertheless, there have been a limited number of studies utilizing feature analysis on rectal MRI to identify the noninvasive independent predictors of tumor stage and positive nodal status [28]. Furthermore, the independent predictors identified in these studies were derived solely from conventional morphological images, whereas our study encompasses both morphological (T2W) and functional ADC maps, providing a more comprehensive analysis. In this study, for the differentiation of tumor stages, the support vector machine (SVM) model performed best on the T2-W and ADC images. During training with 32 subjects, the SVM model achieved an accuracy of 72.93% and an AUC of 0.7 (CI = 0.6–0.74). During testing with 28 subjects, the SVM model achieved an accuracy of 72% and an AUC of 0.69 (CI = 0.67–0.7). Additionally, the use of radiomic features provided insights into the distribution of tumor stages among the participants. These findings align with earlier reports by Ma et al. [10] and Yin et al. [29], which further support the potential of radiomics analysis in differentiating between the different stages of rectal cancer.

The current gold standard for assessing extramural venous invasion (EMVI) and nodal statuses in rectal cancer is post-surgical pathological examination [30]. However, this approach has certain limitations. First, obtaining the pathological EMVI status necessitates surgical intervention, which can delay critical treatment decisions before surgery. Second, the administration of preoperative neoadjuvant therapy may potentially lead to the underestimation of the EMVI status in subsequent postoperative pathological assessments [31]. In light of these challenges, this study has introduced a novel approach based on radiomics analysis, which has demonstrated remarkable advantages in predicting EMVI compared to those of the traditional MRI methods. The logistic regression model’s performance in identifying EMVI status is noteworthy. The high accuracy and AUC values indicate that radiomic features can be indicative of EMVI, which can guide the clinical decisions. However, the inability to definitively ascertain the status for a portion of participants highlights the complexity of the problem. Nevertheless, the model provides valuable insights into EMVI status in the majority of cases, which can aid in more personalized treatment approaches.

Here, the results have surpassed the findings of a previous study conducted by Brown et al., where the specificity of conventional MRI for EMVI detection was reported as 62% [32]. The improved performance observed in our study can be attributed to the combined diagnostic capabilities of the radiomics signature, conventional MRI analysis, and the incorporation of relevant clinical features. This comprehensive approach not only enhances the accuracy of EMVI prediction, but also has the potential to expedite treatment decisions and reduce the reliance on invasive surgical procedures [30].

This study had two major limitations. The first one was the relatively low number of CRC patients included in the study, which could undermine the possibility of demonstrating classification and treatment differences among many patients. However, despite this limitation, the cases used in the study contained all the radiomic features that demonstrated usefulness in the classification of patients according to their KRAS status, tumor staging, or EMVI. The second limitation was that the study adopted a single-center design. The use of a single-center design minimized the number of patients that could be enrolled in the study and increased the risk of selection bias [33]. The single source of participants from one study center also undermined the possibility of making a clinically significant observation of the usefulness of radiomic features. Future studies could address this limitation by adopting a multi-center approach that would increase the sample size and enhance the variability of the participant population.

## 5. Conclusions

In this study, we have successfully employed machine learning models that utilize radiomics data extracted from both ADC maps and T2-weighted images. These models have demonstrated their effectiveness in several critical aspects of colorectal cancer management, including distinguishing between KRAS gene statuses, assessing tumor grading, and determining MRF invasion by identifying extramural venous invasion. These findings highlight the tremendous potential of radiomics as a valuable tool in the realm of precision medicine and clinical decision making. A noteworthy aspect of our study is that the variability of scanners used did not adversely impact the performance of our machine learning models. This resilience to scanner variations suggests that our model is adaptable and generalizable, making it applicable in diverse clinical settings. However, it is imperative to emphasize the importance of standardized imaging protocols in the field of radiomics. The adoption of standardized imaging protocols is critical as it ensures consistency and uniformity in data acquisition. This, in turn, significantly enhances the reliability and reproducibility of MRI radiomics across various clinical contexts. As radiomics continues to assume a prominent role in the diagnosis and management of colorectal cancer, the need for standardization becomes even more pronounced, underlining its importance in maintaining high-quality, consistent results across different healthcare settings.

## Figures and Tables

**Figure 1 diagnostics-13-03541-f001:**
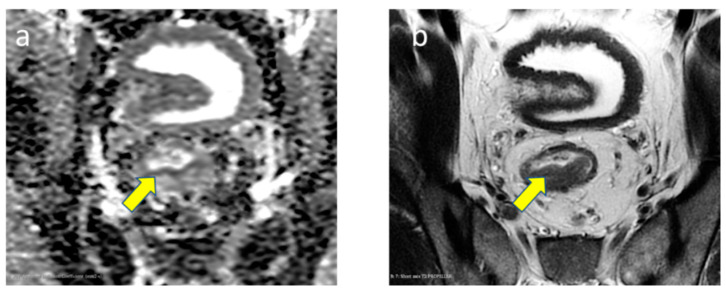
Typical MR imaging appearance of KARAS + or − tumor (yellow arrow) on (**a**) ADC and (**b**) T2-weighted images.

**Figure 2 diagnostics-13-03541-f002:**
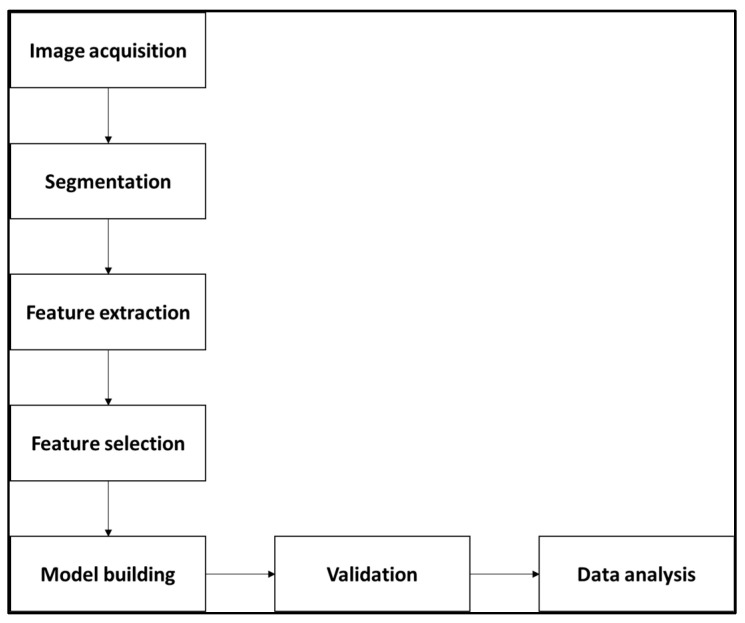
Flowchart illustrating the basic steps of radiomics workflow.

**Figure 3 diagnostics-13-03541-f003:**
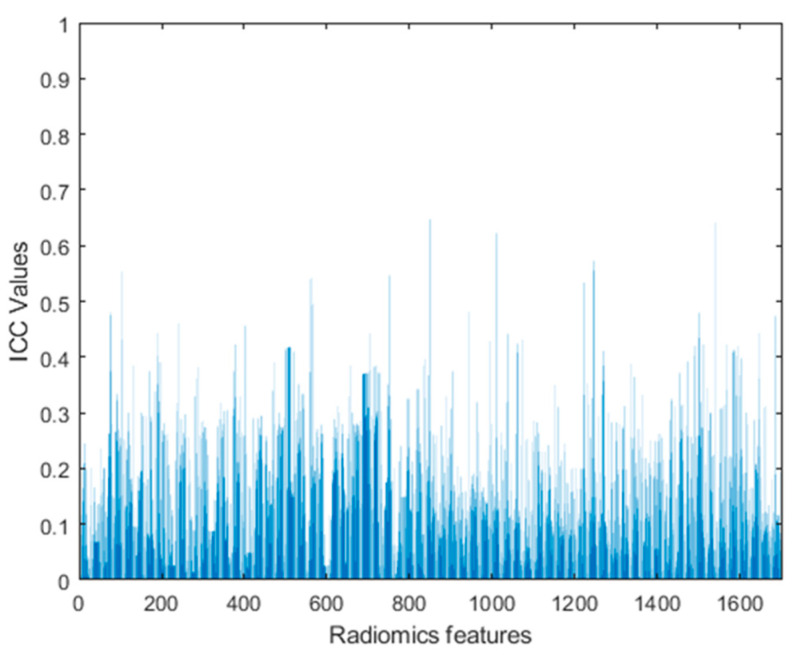
Shows ICC values of radiomic features between two platforms.

**Figure 4 diagnostics-13-03541-f004:**
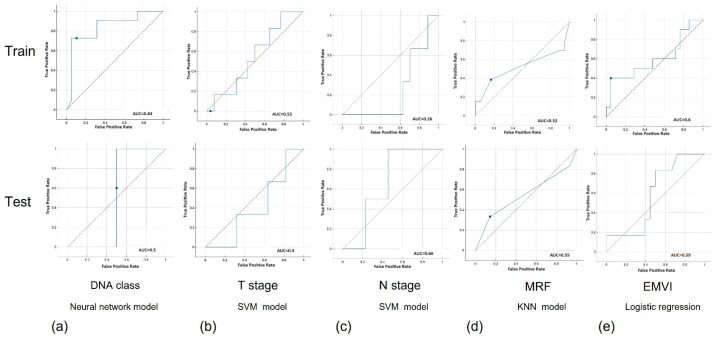
(**a**) ROC curves of train and test to predict DNA. (**b**) ROC curves of train and test to predict t stage. (**c**) ROC curves of train and test to predict N stage. (**d**) ROC curves of train and test to predict MRF. (**e**) ROC curves of train and test to predict EMVI.

**Table 1 diagnostics-13-03541-t001:** Detailed sequence parameters used in T2-weighted and diffusion-weighted imaging (DWI) on both MRI platforms.

Parameters	Platform A		Platform A	
	T2W	DWI	T2W	DWI
Image orientations	Axial	Axial	Axial	Axial
Field of view (mm)	480–500	400–370	480–500	420–380
Matrix	564 × 260	160 × 200	564 × 260	150 × 180
Slice thickness (mm)	5	10	5	10
ETL or FA	Min	90 FA	Min	90 FA
TR (ms)	4000	8100	4200	8300
TE (ms)	85	67	95	73
b values (s/mm^2^)	0–1000–1500		0–800–1200	
Number of Averaging (NSA)	1	1	1	1

**Table 2 diagnostics-13-03541-t002:** Demographic and clinical characteristics of the KRAS-mut and KRAS-wild populations.

	Overall	KRAS-Wild	KRAS-Mut	*p* Value
Patients	60	32	28	
Sex (%)				
male	35 (58.3)	15	20	0.0002
female	25 (41.7)	13	12	>0.9999
Age	57.3 ± 12.9	60.2 ± 19.3	56.5 ± 16.7	0.1392
Weight (kg)	75.6 ± 17.6	71.5 ± 17.7	79.8 ± 16.5	0.0934
Tumor volume (cm^3^)	11.8 ± 16.1	19.7 ± 8.8	14.6 ± 22.4	0.1363
cT stage (%)				
cT1	0 (0.0)	0 (0.0)	0 (0.0)	
cT2	7 (13.5)	5 (71.5)	2 (28.5)	0.8666
cT3	22 (42.3)	14 (63.6)	8 (36.4)	0.0197
cT4	23 (44.2)	13 (56.5)	10 (43.5)	0.8666
cN stage (%)				
cN0	8 (13.8)	4 (50)	4 (50)	>0.9999
cN1	8 (13.8)	2 (25)	6 (75)	0.4245
cN2	42 (72.4)	26 (61.9)	16 (38.1)	<0.0001
cN3	0 (0.0)	0 (0.0)	0 (0.0)	
EMV1				
Negative	18 (25.7)	11 (61.1)	7 (38.9)	0.4245
Positive	28 (54.3)	21 (56.3)	17 (43.7)	0.4245
Unknown	14 (20)	6 (42.9)	8 (57.1)	0.8710
MRF				
Negative	34 (56.7)	14 (41.2)	20 (59.8)	0.0197
Positive	22 (36.7)	10 (45.5)	12 (54.5)	0.9977
Unknown	4 (6.7)	3 (75)	1 (25)	0.6321

**Table 3 diagnostics-13-03541-t003:** Ml classifier performance. Performance metrics of the ML models classified with 95% confidence intervals (CI) and Area Under the Curve (AUC). * *p* < 0.05 training vs. Testing (Accuracy); ^†^
*p* < 0.05 training vs. Testing (AUC).

Category	ML Model	Training ACCURACY%	Training AUC, CI	Testing ACCURACY%	Testing AUC, CI
KRAS+/− class	Neural network	72.93 ± 4.24	0.71 ± 0.08	62.48 ± 8 *	0.68 ± 0.25
TNM stage	SVM	77.6 ± 1.26	0.67 ± 0.062	72 ± 0.02	0.69 ± 0.018 ^†^
MRF (positive/negative)	KNN	62.9 ± 1.75	0.55 ± 0.03	65.83 ± 0.10	0.578 ± 0.055
EMVI (positive/negative)	Logistic regression	73.1 ± 3.33	0.62 ± 0.05	76 ± 1.75	0.595 ± 0.063

## Data Availability

Data is contained within the article or Appendix A. The data presented in this study are available in Appendix A.

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
