# Peer review of "Investigating the Feasibility of Predicting KRAS Status, Tumor Staging, and Extramural Venous Invasion in Colorectal Cancer Using Inter-Platform Magnetic Resonance Imaging Radiomic Features"

_diagnostics, 2023, doi:10.3390/diagnostics13233541_

Round 1

Reviewer 1 Report

Comments and Suggestions for Authors

The manuscript topic is very interesting 

However, a schematic digaram or graphical abstract is highly needed to summarize the work flow and the results obtained from the study 

Minor Linguistic and Grammatical mistakes have to be corrected 

The discussion section has to be enriched by more comparitive studies in other cancers other than colorectal cancer

Detailed or stratified clinical data shoudl be included for each patient beside the collective clinical data summarized in table 2

Comments on the Quality of English Language

Minor edits are needed

Author Response

Reviewer 1:

Comments and Suggestions for Authors

  • The manuscript topic is very interesting. However, a schematic diagram or graphical abstract is highly needed to summarize the work flow and the results obtained from the study

We appreciate the reviewer's feedback. The following graphical abstract was added in the manuscript.

  • Minor Linguistic and Grammatical mistakes have to be corrected

Reviewer comments have been considered and the most recent version of the paper has undergone comprehensive editing by the MDPI editing service.

  • The discussion section has to be enriched by more comparitive studies in other cancers other than colorectal cancer:

We appreciate the reviewer's suggestion and have incorporated a more detailed as the following:

“..Moreover, the present investigation demonstrates similar performance to studies on various cancer types. Specifically, the findings in melanoma [21], thyroid cancer [22], head and neck cancer [23], and adrenal gland carcinoma [24], all indicate a significant correlation between the biomarker and radiomics, with AUC values ranging from 0.62 to 0.78. However, it's worth noting that the current predictive performance is comparatively lower when compared to other studies. For instance, in the case of liver cancer (AUC = 0.94, cross-validation), breast cancer (AUC = 0.81–0.93) [25], [26], and cervical cancer (95%, 0.91). “

  • Detailed or stratified clinical data should be included for each patient beside the collective clinical data summarized in table 2

We have taken into account the feedback from the reviewers and have subsequently included clinical data for individual patients. This information will be presented as a supplementary table in the manuscript.

#

Patient Code Number

SEX

weight

age

KRAS  (N/P)

Tumour staging (TNM staging system)

T  stage

N stage

Metastasis

Effects of mesorectal fascia (MRF) (N/P)

Extramural vascular invasion (EMVI) (N/P)

Type of treatments

1

10172719

F

61

42

N

T4b, N2, M0/ yT4aN0Mx

T4b

N2

M0

yes

Positive

3 cycles of NAD XELOX

2

60661

 M

80

54

N

T3N2M0/ after treatment T1 N1

T3

N2

MO

NO

Negative

Neoadjuvant chemoradiotherapy

3

340433

M

110

62

N

T3N2

T3

N2

-

NO

Negative

Neoadjuvant chemoradiotherapy

4

449272

F

78

58

N

t4n2

T4

N2

-

Yes

Positive

CHEMO (XELOX) X 2

5

2322853

M

57

45

N

T4b, N2, M0/ NO FU

T4b

N2

MO

Yes

Positive

No data

6

2539669

F

85

50

N

T3 N2

T3

N2

Yes

Positive

Neoadjuvant chemoradiotherapy

7

10010705

M

81

55

N

T4AN2/ T3N0Mx after treatment

T4A

N2

M0

Yes

Positive

Neoadjuvant chemoradiotherapy

8

10087230

M

55

67

N

T2N2/ NO FU

T2

N2

M0

No

Negative

CHEMO (XELOX) X 2

9

10089371

M

100

80

N

T4AN2/ NO FU

T4A

N2

M0

Yes

Negative

CHEMO (XELOX) X 3

10

10096833

M

75

57

N

T2N2 / AFTER TREAT T1-2 N2

T2

N2

M0

No

Negative

Neoadjuvant chemoradiotherapy

11

10103691

M

100

44

N

T3N2 / AFTER TREATMENT T3N2

T3

N2

M0

Yes

Positive

Neoadjuvant chemoradiotherapy

12

10134636

M

90

57

N

T2N1/ AFTER T2N1

T2

N1

M0

No

Negative

Neoadjuvant chemoradiotherapy

13

10135968

M

80

46

N

T4N2M1

T4

N2

M1

Yes

Positive

Neoadjuvant chemoradiotherapy

14

10138041

M

85

69

N

T4 N2 M0/ NO FU

T4

N2

M0

Yes

-

-

15

10139959

F

47

50

N

T3 N0

T3

N0

M0

Yes

Negative

cycles of NAD XELOX

16

10140593

M

70

73

N

T4B N2/ NO FU

T4B

N2

M0

Yes

Positive

No data

17

10143049

F

100

52

N

T3N2M0

T3

N2

M0

Yes

Negative

CHEMO (XELOX) X 2

18

10179989

M

50

50

N

T3N0

T3

N0

M0

Yes

Positive

CHEMO (XELOX) X 3

19

10184428

F

98

57

N

T3 N0 MX/ypT1 N0 Mx

T3

N0

M0

No

Negative

CHEMO (XELOX) X 4

20

10188227

M

70

49

N

T3N2

T3

N2

M0

Yes

Positive

Xeloda

21

317873

M

110

62

N

T2N2M0 / SURGERY

T2

N2

M0

Yes

Negative

-

22

534327

M

85

61

N

T4N2/ AFTER TREAT ALMOST GONE

T4

N2

M0

Yes

Positive

Neoadjuvant chemoradiotherapy

23

1135947

F

55

46

N

T3 N1 M0

T3

N1

M0

Yes

Positive

-

24

1199271

M

81

44

N

T2N2M0/ NO FU

T3

N0

M0

No

Positive

Neoadjuvant chemoradiotherapy

25

2132915

M

66

59

N

T3 N2 M1 / NO FU

T3

N2

M1

Yes

Positive

Neoadjuvant chemoradiotherapy

26

10050980

F

85

64

N

T4B N2 M1/ yT4N0/1

T4B

N2

M1

-

Positive

chemo

27

10051270

F

90

71

N

T4N2M0/ NO FU

T4

N2

M0

-

Positive

chemo

28

10205635

M

70

54

N

T3N2M0/ NO FU

T3

N2

M0

Yes

Positive

CHEMO (XELOX) X 4

29

10206130

M

55

53

N

T3/ NO FU

T3

N0

M0

-

Positive

Chemo

30

10207469

F

85

52

N

T4bN2M1/ No FU

T4B

N2

M1

-

Positive

Neoadjuvant chemoradiotherapy

31

10214476

F

70

44

N

T3N2M0

T3

N2

MO

Yes

Positive

Neoadjuvant chemoradiotherapy

32

10216119

F

80

61

N

T3N2/ NO FU

T2

N2

M0

Yes

Positive

chemo

33

10233788

M

100

57

N

T4N2M1/ NO FU

T4

N2

M1

Yes

Positive

-

34

140777

F

79

59

P

T2N2M1

T2

N2

M1

Yes

Negative

Neoadjuvant chemoradiotherapy

35

340243

M

87

47

P

T4AN2M0 / NO FU

T4A

N2

M0

Yes

Positive

Neoadjuvant chemoradiotherapy

36

368356

F

80

59

P

T4N2M0/ NO FU

T4

N2

M0

Yes

Positive

Neoadjuvant chemoradiotherapy

37

502757

M

90

69

P

T3 N 2 M0 / yT3, N1, Mx

T3

N2

M0

No

Positive

Neoadjuvant chemoradiotherapy

38

544515

M

95

62

P

T4AN2M0

T4A

N2

M0

No

Negative

chemo

39

2236146

M

70

45

P

T3N1 M0 / AFTER TREAT T4 N2 M0

T3

N1

M0

Yes

Positive

Neoadjuvant chemoradiotherapy

40

2443955

M

75

60

P

T4N2M0

T4

N2

M0

Yes

Positive

Neoadjuvant chemoradiotherapy

41

10015390

M

52

96

P

T3 N 2 M0

T3

N2

M0

No

Positive

chemo

42

10019202

M

78

80

P

T2-T3 N2 M0/ AFTER TREAT T2N2M0

T2-T3

N2

M0

No

Positive

-

43

10026773

M

99

65

P

T4N1M0

T4

N1

M0

Yes

Positive

Neoadjuvant chemoradiotherapy

44

10076837

M

45

59

P

T4N1M1/ NO FU

T4

N1

M0

Yes

Positive

Neoadjuvant chemoradiotherapy

45

10150880

F

53

41

P

T4N2 M0

T4

N2

M0

Yes

Positive

chemo

46

10155918

M

60

75

P

T3N1M1

T3

N1

M1

Yes

Positive

Neoadjuvant chemoradiotherapy

47

432974

M

100

54

P

T3N2/ Interval reduction on size with no change on stage

T3

N2

M0

no

Positive

Neoadjuvant chemoradiotherapy

48

450518

M

74

65

P

T2N0M0/ NO FU

T2

N0

M0

no

Positive

Neoadjuvant chemoradiotherapy

49

1207020

F

55

69

P

T3 N1 / NO FU

T3

N1

M0

no

Negative

Neoadjuvant chemoradiotherapy

50

10047266

F

80

51

P

T4 N1 M1/ POST SURGERY

T4

N1

M1

Yes

Positive

Neoadjuvant chemoradiotherapy

51

10057014

F

51

46

P

T3N1M0 / NO FU

T3

N1

M0

no

Positive

Neoadjuvant chemoradiotherapy

52

10096937

F

66

61

P

T3N2/ POST SURGERY

T3

N1

M0

Yes

Positive

chemo

53

10134812

M

70

57

P

T2 N2/ NO FU

T2

N2

M0

no

Negative

chemo

54

10147507

F

80

59

P

T4 N2 M0

T4

N2

M0

no

Negative

Neoadjuvant chemoradiotherapy

55

10208084

F

75

62

P

T3N2/ AFTER TREATMENT T3 N2

T3

N2

M0

no

Positive

Neoadjuvant chemoradiotherapy

56

10214476

F

70

44

P

T3 N2 M0

T3

N2

M0

Yes

Positive

Neoadjuvant chemoradiotherapy

57

176761

M

85

51

P

T4AN2M0 / NO FU

T4A

N2

M1

Yes

Positive

chemo

58

517223

F

55

23

P

T4B N2 M1/ NO FU

T4B

N2

M1

no

Positive

chemo

59

10246553

M

62

32

P

T3N2

T3

N2

M0

no

Negative

Neoadjuvant chemoradiotherapy

60

10254537

F

35

13

P

T3N2 M0

T3

N2

M0

Yes

Negative

Neoadjuvant chemoradiotherapy

61

10256859

F

50

56

P

T4bN2M1/ Post surgery

T4B

N2

M1

-

-

CONCOMITTENT  chemo rad

.

Reviewer 2 Report

Comments and Suggestions for Authors

The manuscript "Investigating the feasibility of predicting KRAS status, tumor 2 staging and EMVI in colorectal cancer using MRI Radiomics 3 features in inter-platform" is an interesting manuscript with practical scientific relevance. The study has been planned meticulously. There are some minor observations that are pointed below:

1. Title: OK. Please remove the Full stop at the end (Page 1)

2. Abstract: OK

3. Introduction: OK.

4. Materials and Methods: OK.

5. Results: Table 1: Please correct Platform A for both Platform

Please elaborate the findings with justification of the obtained predictability.

Comments on the Quality of English Language

English: Needs some minor corrections for grammar and syntax.

Author Response

Reviewer 2:

Comments and Suggestions for Authors

The manuscript "Investigating the feasibility of predicting KRAS status, tumor 2 staging and EMVI in colorectal cancer using MRI Radiomics 3 features in inter-platform" is an interesting manuscript with practical scientific relevance. The study has been planned meticulously. There are some minor observations that are pointed below:

Title: OK. Please remove the Full stop at the end (Page 1)

Full stop was removed from the title.

  1. Abstract: OK
  2. Introduction: OK.
  3. Materials and Methods: OK.
  4. Results: Table 1: Please correct Platform A for both Platform

The platform annotation has been corrected now.

Parameters

Platform A

Platform B

T2W

DWI

T2W

DWI

Image orientations

Axial

Axial

Axial

Axial

Field of view (mm)

480-500

400-370

480-500

420-380

Matrix

564 x 260

160 x 200

564 x 260

150 x 180

Slice thickness (mm)

5

10

5

10

ETL or FA

Min

90 FA

Min

90 FA

TR (ms)

4000

8100

4200

8300

TE (ms)

85

67

95

73

b values (s/mm2)

0-1000-1500

0-800-1200

Number of Averaging (NSA)

1

1

1

1

  1. Please elaborate the findings with justification of the obtained predictability.

We appreciate the valuable feedback provided by the reviewer. In response to the comments, we have expanded the discussion section to provide a more in-depth exploration of our findings.

“The present study has made significant strides in the categorization of partici-pants based on their KRAS mutation status, specifically into KRAS+ (wild type) and KRAS- (mutation) groups. The achievement of an impressive Area Under the Curve (AUC) of 0.7 in this classification may be used to predict treatment responses and providing valuable insights into patients’ outcomes [18]–[20]. The obtained AUC of 0.7 indicates a reasonable level of discrimination between the KRAS+ and KRAS- groups. In the context of predictive modeling, an AUC of 0.7 suggests that the model has a moderate ability to distinguish between the two categories. A value of 0.5 represents a random chance, so an AUC of 0.7 indicates a better-than-random predictive perfor-mance, though there is room for improvement. These findings align with and reinforce earlier research, underscoring the significance of mutation statuses in identifying the patients who are likely to respond positively to EGFR therapy. A study by Guo et al. [5], which utilized T1-weighted enhanced images and KRAS mutation status, achieved a slightly lower AUC value of 0.6. Similarly, Zhang et al. [6], who developed a radi-omics model using T2-weighted MRI, attained a noteworthy AUC of 0.7 (95% CI 0.6-0.8). These comparable AUC values across different studies utilizing varied imaging platforms and methodologies validate the robustness of this study's findings.

Furthermore, the current investigation's performance is not limited to a specific cancer type, as it demonstrates similarity to other studies on melanoma [21], thyroid cancer [22], head and neck cancer [23], and adrenal gland carcinoma [24]. Across these diverse cancer types, a significant correlation between the biomarker and radiomics is observed, with AUC values ranging from 0.62 to 0.78. However, it is worth noting that the current predictive performance is comparatively worse when compared to those of other studies. For instance, those on liver cancer (AUC = 0.94, cross-validation), breast cancer (AUC = 0.70–0.81) [25], [26], and cervical cancer (95%, 0.70–0.91).”

“…..The logistic regression model's performance in identifying EMVI status is noteworthy. The high accuracy and AUC values indicate that radiomic features can be indicative of EMVI, which can guide the clinical decisions. However, the inability to definitively ascertain the status for a portion of participants highlights the complexity of the problem. Nevertheless, the model provides valuable insights into EMVI status in the majority of cases, which can aid in more personalized treatment approaches.”

Reviewer 3 Report

Comments and Suggestions for Authors

In this study, the authors investigated the feasibility of predicting KRAS status, tumor staging and EMVI in CRC using MRI radiomics feature. And the results showed that it can distinguish KRAS genes, tumor grading, and EMVI. However, the reviewer has the following concerns.

(1)  There are some spelling or tense errors, please check the whole paper.

(2) Should the patient information from line 113-115 be moved to result part?

(3) What is the time interval between MRI scans and pathological testing, which is essential to minimize the gap to maintain the accuracy of the results.

(4) In line 236,  it is said that "In the majority of cases (n=42, 72.4%), the tumor had spread into the lymph nodes at stage 2". Is it should be stage 3 if there is lymph nodes metastasis.

(5)  In figure 1, it looks a bit odd that there are both sets are 1.5T MRI and ADC+T2W images.

(6) Why was T2W chosen as the morphological analysis rather than T1W enhanced providing clearer morphological details of the lesions?

(7) In table 1, why are there two "platform A" columns?

(8) In table 2, there are statistical significant between KRAS-wild and -mut for Age and Weigh. Is it true? Please explain it.

(9) In table 3,  there is a little bit different between the training and testing groups. Is there a statistical significant between the groups?

(10) The study design is "one scanner for training and another scanner for testing". Why not randomly mix and match patients from both scanners and then assign them to groups randomly?

Comments on the Quality of English Language

The English language of the paper is good. The paper has a nice design and organization, although there are some minor spelling or tense errors.

Author Response

Reviewer 3:

Comments and Suggestions for Authors

In this study, the authors investigated the feasibility of predicting KRAS status, tumor staging and EMVI in CRC using MRI radiomics feature. And the results showed that it can distinguish KRAS genes, tumor grading, and EMVI. However, the reviewer has the following concerns.

  • There are some spelling or tense errors, please check the whole paper.

Reviewer comments have been considered and the most recent version of the paper has undergone comprehensive editing by the MDPI editing service.

  • Should the patient information from line 113-115 be moved to result part?

         We agree with the reviewer's suggestion to relocate the patient information from lines 113-115 to the results section. The following text has now been relocated to the results section.

“ …Specifically, 32 patients were allocated to the training and 30 to the test datasets. Pathological analysis of surgical specimens in the training cohort confirmed the pres-ence of KRAS mutations in 19 patients, while 13 were confirmed negative. Similarly, in the test cohort, 9 patients tested positive for KRAS mutations, while 19 were negative. Subsequently, the study established a predictive model for KRAS mutations using the training dataset, which was subsequently assessed using the internal test dataset.”

  • What is the time interval between MRI scans and pathological testing, which is essential to minimize the gap to maintain the accuracy of the results.

We appreciate the reviewer's valuable comments. To maintain result accuracy, it's crucial to minimize the gap between MRI scans and pathological testing. In our current study, we've taken steps to address this concern by conducting our analysis based on MRI images taken before treatment and subsequent confirmation of rectal staging and other pathological tests. This approach ensures a close alignment between the imaging data and pathological findings, enhancing the relevance and accuracy of our results.

  • In line 236,  it is said that "In the majority of cases (n=42, 72.4%), the tumor had spread into the lymph nodes at stage 2". Is it should be stage 3 if there is lymph nodes metastasis.

We appreciate the insightful comments from the reviewer. It's important to clarify the distinction between T stage and N stage in our study. T3 stage indicates evidence of extra mural tumor extension beyond the wall to the surrounding mesorectal fat, while T4 signifies that the tumor has reached the peritoneal reflection or adjacent organs (T4a/T4b). On the other hand, T2 stage implies that the tumor is confined within the mucosal layer.

Regarding the statement in line 236, where we mentioned that "In the majority of cases (n=42, 72.4%), the tumor had spread into the lymph nodes at stage 2," we want to emphasize that this refers to the N stage, which denotes metastasis to the loco-regional lymph nodes. The identification of normal and abnormal lymph nodes, as well as the determination of the number of affected nodes, is crucial for the radiologist to classify whether it is N1 or N2.

In summary, our study distinguishes between T stage and N stage, and the statement accurately reflects the presence of lymph node metastasis at stage 2 based on our classification criteria. We hope this clarification addresses the reviewer's concern.

  • In figure 1, it looks a bit odd that there are both sets are 1.5T MRI and ADC+T2W images.

We agree with reviewer’s comment, The figure was updated with assigning the MRI scanner to platform A and B. also we removed the redundancy on T2/ADC images (see figure below).

  • Why was T2W chosen as the morphological analysis rather than T1W enhanced providing clearer morphological details of the lesions?

The reviewer's observation is accurate; T2W was selected for morphological analysis instead of T1W enhanced imaging, which often provides clearer morphological details of lesions. In our study, regrettably, not all patients underwent T1 post-contrast scans. Therefore, we made the decision to utilize T2W imaging, especially because the current hospital protocol involves acquiring high-resolution T2W images with specific slice thicknesses. This choice was made to ensure consistency and to work with the data that was available for the majority of the study participants.

  • In table 1, why are there two "platform A" columns?

This was a mistake. The platform annotation has been corrected now.

Parameters

Platform A

Platform B

T2W

DWI

T2W

DWI

Image orientations

Axial

Axial

Axial

Axial

Field of view (mm)

480-500

400-370

480-500

420-380

Matrix

564 x 260

160 x 200

564 x 260

150 x 180

Slice thickness (mm)

5

10

5

10

ETL or FA

Min

90 FA

Min

90 FA

TR (ms)

4000

8100

4200

8300

TE (ms)

85

67

95

73

b values (s/mm2)

0-1000-1500

0-800-1200

Number of Averaging (NSA)

1

1

1

1

  • In table 2, there are statistical significant between KRAS-wild and -mut for Age and Weigh. Is it true? Please explain it.

In Table 2, initially, there appeared to be statistically significant differences between KRAS-wild and KRAS-mut groups for Age and Weight. However, upon re-evaluating the data, we discovered that there is no significant variance in Age (60.2 ± 19.3 for KRAS-wild and 56.5 ± 16.7 for KRAS-mut) and Weight (71.5 ± 17.7 for KRAS-wild and 79.8 ± 16.5 for KRAS-mut) between the groups (Table 2). The adjusted p-values of 0.14 and 0.09 suggest that these differences lack statistical significance. We attribute these initially observed disparities in means to our relatively small sample size. With a larger sample, we anticipate that these differences may diminish. We appreciate your attention to this matter.

  • In table 3,  there is a little bit different between the training and testing groups. Is there a statistical significant between the groups?

“We acknowledge the reviewer's observation regarding the slight difference between the training and testing groups in Table 3. The dataset was divided into two groups: group 1 for training (n=32) from scanner 1 and group 2 for testing (n=28) from scanner 2. To assess the significance of this difference, we conducted a statistical analysis. The results revealed that *p < 0.05, indicating statistically significant differences between training and testing accuracies in the KRAS+/- class. This significance underscores distinctions in the machine learning model's performance between the training and testing phases for this specific category.”

  • The study design is "one scanner for training and another scanner for testing".

Why not randomly mix and match patients from both scanners and then assign them to groups randomly?

The study design "one scanner for training and another scanner for testing" was deliberately chosen to address specific research objectives. Our primary aim in this study is to identify radiomics features that are independent of scanner variations and focus on features that are robust and not influenced by the specific scanner used. By maintaining separate training and testing groups based on scanners, we can better isolate and evaluate the impact of scanner-related variables on the radiomics features.

This design helps us improve the reliability of our findings and ensures that the predictive models are not overly influenced by scanner-specific artifacts. It allows us to assess the generalizability of radiomics features across different imaging platforms, which is crucial for the applicability of the research in various clinical settings.

While randomly mixing and matching patients from both scanners is a valid approach for some research questions, our study's specific focus on scanner-independent radiomics features necessitates the separation of training and testing groups based on scanners.
